# OpenReview forum: "Safety Mirage: How Spurious Correlations Undermine VLM Safety Fine-Tuning and Can Be Mitigated by Machine Unlearning"
_ICLR.cc/2026/Conference — ICLR 2026 Poster_

### Official Review · Reviewer_Gye9 · 2025-10-28

**Soundness:** 2
**Presentation:** 3
**Contribution:** 2
**Rating:** 4
**Confidence:** 3

**Summary:**

The paper uncovers a “safety mirage” in safety fine-tuning of MLLM, which is caused by spurious correlations between questions and safety labels in training data. On the attack side, the authors show these models are still jailbreakable via a simple one-word substitution that exploits these correlations, and they link the over-prudence (benign refusals) to the same bias. On the defense side, they propose machine unlearning to remove the influence of unsafe responses without relying on spurious shortcuts. Extensive experiments across multiple safety benchmarks and VQA tasks validate the mirage phenomenon and show that MU-based fine-tuning mitigates spurious correlations and reduces over-prudence while preserving utility.

**Strengths:**

1. This paper identifies an important issue in safety fine-tuning of MLLM: spurious correlation between rejecting/answering behaviours and certain keywords in the query. The authors demonstrate this clearly through investigating the word frequency in the training dataset and one-word jailbreak/over-prudence experiment.

2. The adopted unlearning objectives achieves better safety and mitigates over-prudence

**Weaknesses:**

1.Lack of explanation and motivations for the unlearning objectives. The authors are doing a good job identifying the problem of spurious correlation certain reject/answer behaviours and certain key words in the query. However, it is unclear why the use of machine unlearning objectives could mitigate the spurious correlation problem.

2. Without a clear motivation of the unlearning objectives, the reviewer can only link the effect of unlearning to regularization. The spurious correlation might result from the MLLM overfitting to the safety fine-tuning datasets. What if the authors apply certain regularization techniques such as lager weight decay, smaller lr or less training steps? In addition, data augmentation strategy that expands the query space might also be useful.

3. Possible side-effects of the unlearning objectives. The adopted RMU intends to erase the model's representation of unsafe queries/responses. However, it is unclear how would the fine-tuned model behave when seeing similar queries. Would the model output nothing or generate irrelevant texts?

**Questions:**

1. What are the inputs to the MLLM when conducting the K-shot attack in Figures 3, 4 and in other evaluation? Are the image paired with K versions of the paraphrased queries or paired with one of the version at a time? Why do the authors choose K-shot attack throughout this paper?

2. Can the authors provide examples of one-word jailbreaking and over-prudence? I.e., how is the keywords are integrated in to the original query.

3. When performing NPO, how are the loss mask set? In normal instruction tuning, we would mask the prompt/instruction tokens and only compute loss on the response tokens. In the case of NPO, if only the queries were provided (lines 342-343), would the prompt tokens be masked?

---

> ### Author Response · Authors · 2025-11-21
> **Response to Reviewer Gye9 (Part I)**
>
> We thank the reviewer for the thoughtful and constructive feedback.
> We appreciate your positive recognition that: our paper “identifies an important issue in safety fine-tuning”, and that our proposed MU-based objectives “achieve better safety and mitigate over-prudence.”
>
> Below, we address each weakness and question in turn, denoted by **[W]** for weaknesses and **[Q]** for questions.
>
> **[W1]**: Lack of explanation and motivations for the unlearning objectives. The authors are doing a good job identifying the problem of spurious correlation certain reject/answer behaviours and certain key words in the query. However, it is unclear why the use of machine unlearning objectives could mitigate the spurious correlation problem.
>
> **Response:**
> Thank you for raising this point. The motivation for MU follows directly from our Section 4 findings: safety fine-tuning introduces **label-driven spurious correlations** between superficial textual cues (e.g., question word choice) and the reject label. Because these correlations arise *from the labels themselves*, a **label-free mitigation strategy** is particularly well suited for this problem. As Reviewer kJt9 also noted, “the use of machine unlearning techniques to address the spurious correlation problem is well-motivated, as they are label-free.”
>
> MU explicitly removes the influence of harmful data and representations, rather than relying on supervised fine-tuning over surface-level data. This allows MU to weaken the spurious correlation that SFT inadvertently reinforces. **Specifically**, NPO pushes the model away from the reference model’s behavior on unsafe queries, reducing its dependence on spurious textual cues. RMU randomizes internal representations associated with unsafe samples, directly breaking the mapping between superficial cues (e.g., first-word choice) and the model’s output. Importantly, both mechanisms operate without using safety labels, preventing the model from re-forming the same shortcut correlations that arise in SFT-trained models.
>
> We hope that the above explanation of the unlearning objectives clarifies why unlearning is a principled and convincing approach for mitigating spurious correlations.
>
> **[W2]**: Without a clear motivation of the unlearning objectives, the reviewer can only link the effect of unlearning to regularization. The spurious correlation might result from the MLLM overfitting to the safety fine-tuning datasets. What if the authors apply certain regularization techniques such as larger weight decay, smaller lr or fewer training steps? In addition, data augmentation strategy that expands the query space might also be useful.
>
> **Response:**
> Thank you for the thoughtful question. Below we clarify why neither regularization nor data augmentation can replace MU.
>
> - **The spurious correlation is not due to simple overfitting.**
> Our analysis is performed on strong, publicly released safety-fine-tuned models that are already considered robust in the existing VLM safety benchmarks [R1–R3]. Yet these models still fail under the one-word attack. This indicates that the shortcut arises from the safety-label–driven fine-tuning paradigm itself, rather than overfitting to a particular dataset.
>
> - **The one-word attack directly targets the root cause identified in Section 4.**
>   We show that specific first-word patterns are statistically entangled with safety labels in existing datasets. The one-word attack is effective because it aligns with this correlation, providing causal evidence of the shortcut instead of relying on arbitrary perturbations.
>
> - **MU is fundamentally different from standard regularization.**
>   Techniques such as weight decay, smaller learning rates, or fewer training steps globally restrict model capacity but cannot selectively break the shortcut linking first-word cues to refusal behavior. MU, in contrast, is label-free and applies directional, asymmetric updates (NPO/RMU) that specifically weaken shortcut-driven refusals while preserving general capability.
>
> - **Data augmentation is costly, hard to control, and may simply shift shortcuts.**
>   LLM-based paraphrasing requires significant computation and often human verification, especially for multimodal data. It may also introduce new linguistic biases, and because all paraphrases may still receive the same supervision signal (e.g., “reject” as the safety label), shortcuts can reappear in different forms. MU avoids these issues by being dataset-agnostic and not dependent on modifying or expanding the dataset.

---

> ### Author Response · Authors · 2025-11-21
> **Response to Reviewer Gye9 (Part II)**
>
> **[W3]** Possible side-effects of the unlearning objectives. The adopted RMU intends to erase the model's representation of unsafe queries/responses. However, it is unclear how the fine-tuned model would behave when seeing similar queries. Would the model output nothing or generate irrelevant texts?
>
> **Response:**
> Thank you for raising this concern. We want to emphasize that MU is applied solely to remove the influence of harmful knowledge from training data in VLMs, **not** to degrade utility. In fact, its objective is to preserve the model’s general utility (Eq. (4)) while mitigating the unsafe behaviors encoded in the forget set $D_u$.
>
> To assess whether MU causes undesirable behaviors like empty outputs or irrelevant content when encountering similar unsafe queries, we respond across three categories of queries: Unsafe queries, Rephrased unsafe queries, and General benign queries.
>
> - **Unsafe queries (harmful prompts):**
>   Table 2 (Lines 448–464) shows that MU maintains semantic relevance rather than collapsing responses. By decomposing the safety rate (1 – ASR) into IR (irrelevant-but-safe responses) and RR (explicit refusals), we observe: SFT baselines (Mixed-SFT, Post-hoc-SFT) rely almost entirely on high RR, reflecting over-prudence; MU (NPO/RMU) achieves safety mainly via IR responses, avoiding unsafe outputs **without** producing empty text. As illustrated in Fig. 5, RMU-unlearned VLMs produce irrelevant but safe outputs for unsafe queries, rather than collapsing into “no output.”
>
> - **Rephrased unsafe queries (semantics-preserving paraphrases):**
>   Since the K-shot attack is the paraphrased version of the original unsafe query with unchanged semantics, we report safety performance under 1-shot and 3-shot attacks in Table R1 to evaluate MU’s behavior on rephrased unsafe prompts. The **low ASR of MU models** on these paraphrased unsafe queries demonstrates that MU mitigates the spurious correlation rather than overfitting to specific textual forms, while also maintaining general utility.
>
> **Table R1**: Safety (ASR↓) evaluation on VLGuard. “Before” and “After” correspond to the evaluation using and not using one-word attack.
>
> | Models            | Safety Evaluation (ASR, ↓) |   Safety Evaluation (ASR, ↓)  |   Safety Evaluation (ASR, ↓)   |
> |-------------------|-----------------------------|------------------|-----------------------|
> |                   | Before One-Word Attack      | After 1-shot One-Word Attack | After 3-shot One-Word Attack |
> | LLaVA-1.5-7B      | 64.25%                      | 74.43%           | 90.27%                |
> | +Unsafe-Filter    | 65.66%                      | 74.66%           | 90.72%                |
> | + Mixed-SFT       | 0.23%                       | 24.66%           | 54.98%                |
> | + Post-hoc        | 0.23%                       | 25.34%           | 46.83%                |
> | + NPO-Unlearning  | 2.49%                       | 6.99%            | 12.92%                |
> | + RMU-Unlearning  | 1.29%                       | 5.06%            | 10.18%                |
>
> - **General benign queries (utility behavior):**
>   Across standard utility benchmarks (Tables 1 and A1), MU models show **no increase** in undesirable behaviors like blank outputs, maintain stable generation and reasoning capabilities. Moreover, the utility impact is **controllable**. Appendix D.4 (referred in Lines 415–418) shows that coreset unlearning can preserve even higher utility while still reducing ASR and RR, giving MU a flexible safety–utility trade-off that conventional safety fine-tuning cannot provide.
>
> **[Q1]** What are the inputs to the MLLM when conducting the K-shot attack in Figures 3, 4 and in other evaluation? Are the image paired with K versions of the paraphrased queries or paired with one of the versions at a time? Why do the authors choose K-shot attack throughout this paper?
>
> **Response:**
> Thank you for the question.
>
> - **Input format** (how the image and queries are paired):
>   For each image, we generate K paraphrased adversarial queries based on the original question, and we run K separate inference passes. Each pass inputs: the same image, paired with one paraphrased query at a time. The final attack success rate (ASR) is computed by aggregating the K results (e.g., if *any* of the K queries jailbreaks the model, the attack is counted as successful). This setting follows standard K-shot prompting practice in the jailbreak literature [R4, R5].
>
> - **Why we use K-shot evaluation:**
>   We aim to ensure a **stronger and more robust attack setting**. As shown in Fig. 3–4 (Lines 247–283), both ASR and RR results indicate that even 1-shot attacks are already highly effective: a 1-shot one-word attack significantly increases ASR for safety-fine-tuned models.   Since the attack is already strong at K = 1, using a small K (e.g., K = 3) further ensures evaluation under stress-tested adversarial conditions. Notably, K does not need to be large.

---

> ### Author Response · Authors · 2025-11-21
> **Response to Reviewer Gye9 (Part III)**
>
> **[Q2]** Can the authors provide examples of one-word jailbreaking and over-prudence? I.e., how the keywords are integrated into the original query.
>
> **Response:**
> Thank you for your question. Examples illustrating this behavior are already provided in Section 6.2 (Figure 5) of the main paper, as well as Figures A1 and A2 in Appendix A (referenced in Lines 197–201).
>
> **[Q3]** When performing NPO, how are the loss masks set? In normal instruction tuning, we would mask the prompt/instruction tokens and only compute loss on the response tokens. In the case of NPO, if only the queries were provided (lines 342–343), would the prompt tokens be masked?
>
> **Response:**
> Thank you for the question. We believe the confusion arises from the interpretation of Lines 349-350 (original Lines 342–343), and we appreciate the opportunity to clarify:
>
> - **NPO does use the retain set $D_r$.** As stated in Lines 346-347 (original Lines 340–341), the safe query–answer pairs form the retain dataset $D_r$, which is used to compute the retain loss $\ell_r$ in Eq. (4). Therefore, NPO training always includes both: retain loss $\ell_r$ computed on the safe set $D_r$, and unlearning loss $\ell_u$ computed on the unsafe set $D_u$. Lines 349-350 (original Lines 342–343) describe only how $D_u$ is constructed, not the full training data used during NPO.
>
> - **Why unsafe queries alone are sufficient for $D_u$.** The statement “we only use unsafe queries as $D_u$” means that unsafe samples do not require ground-truth answers, because the NPO KL term (Eq. (3)) *compares the model-generated response with the reference model distribution*. Thus, refusals or template-based labels are unnecessary.
>
> - **How masking works in NPO.** NPO masks prompt/instruction tokens exactly as in normal instruction tuning. For the retain loss $\ell_r$, all prompt tokens are masked, and loss is computed only on the ground-truth response tokens. For the unlearning loss $\ell_u$, the model first generates its own response tokens for the unsafe query. The NPO KL-divergence loss is applied **only** to these generated response tokens. Prompt tokens remain masked and do not contribute to $\ell_u$.
>
> > **Reference**
> > **[R1]** Zhang, Yongting, et al. "SPA-VL: A Comprehensive Safety Preference Alignment Dataset for Vision Language Models." CVPR, 2025.
> > **[R2]** Zong, Yongshuo, et al. "Safety fine-tuning at (almost) no cost: A baseline for vision large language models." ICML, 2024.
> > **[R3]** Liu, Xin, et al. "Mm-safetybench: A benchmark for safety evaluation of multimodal large language models." ECCV, 2024.
> > **[R4]** Anil, Cem, et al. "Many-shot jailbreaking." NeurIPS 37, 2024: 129696–129742.
> > **[R5]** Yi, Sibo, et al. "Jailbreak attacks and defenses against large language models: A survey." arXiv:2407.04295, 2024.

---

> > ### Author Response · Authors · 2025-11-25
> > **Look forward to your feedback**
> >
> > Dear Reviewer Gye9,
> >
> > A few days ago, we submitted our responses and have now uploaded a revised version of the paper, with all changes highlighted in blue, including the key points you raised that require special attention.
> >
> > We are writing to kindly check whether you have any follow-up questions or additional comments, or if our responses have sufficiently addressed your concerns. We would be happy to clarify any remaining points and continue the discussion during the open review period. We truly appreciate your time, constructive feedback, and engagement with our work.
> >
> > Authors

---

### Official Review · Reviewer_ZeCK · 2025-10-30

**Soundness:** 3
**Presentation:** 3
**Contribution:** 4
**Rating:** 6
**Confidence:** 5

**Summary:**

This paper explains that existing fine-tuning-based safety alignment methods do not truly enhance a model’s safety capabilities. Instead, they rely on spurious correlations triggered by special tokens to induce refusal behaviors. Building on this observation, the authors propose a one-word jailbreak, showing that simply replacing and repeating a single word before answering can efficiently jailbreak safety-aligned models. To address this issue, the paper introduces a machine unlearning approach and demonstrates its effectiveness using NPO and RMU evaluations.

**Strengths:**

1. The paper is well-organized and clear
2. The content is rich, containing both analysis, jailbreak, and defense of vision-language model safety. The analysis of "spurious correlation" is interesting and insightful
3. The proposed one-word jailbreak and machine unlearning methods are effective and achieve strong performance across various benchmarks

**Weaknesses:**

1. The paper only analyzes the spurious correlation between the first word in the query and the rejection responses. However, this is not the only possible factor, there may also be correlations involving other words in the query that are not in the first position.
2. After unlearning, the utility performance of models degrades more than that of safety fine-tuning methods.
3. All the experiments are conducted on LLaVA-1.5 models, which are relatively old models (released 2 years ago). If the author can provide some analysis based on experiments on a recent model (Qwen series), it will further strengthen the findings and analysis.
4. The current results show that the ASR and utility of the model after unlearning (before jailbreaking) are inferior to those of the safety fine-tuned model. Would it be possible to perform unlearning on the safety fine-tuned model to eliminate the previously analyzed correlations?
5. The “Safety Mirage” described in the paper seems to lack properties specific to vision-language models; it appears to be a phenomenon similar to what has been observed in LLMs. A further discussion on how visual information influences model safety would enhance the completeness

**Questions:**

See Weakness

---

> ### Author Response · Authors · 2025-11-21
> **Response to Reviewer ZeCK (Part I)**
>
> We thank you for your thorough evaluation and insightful comments. We appreciate your positive remarks on the clarity of our paper, the completeness of our analysis–jailbreak–defense pipeline, and the effectiveness of both the one-word jailbreak and our machine unlearning (MU) methods. Your recognition of the spurious-correlation analysis as insightful is particularly encouraging.
>
> Below, we address each of your concerns in detail, denoted by **[W]** for weaknesses and **[Q]** for questions.
>
> **[W1]:** The paper only analyzes the spurious correlation between the first word in the query and the rejection responses. However, this is not the only possible factor—there may also be correlations involving other words in the query that are not in the first position.
>
> **Response**:
> Thank you for your question! We agree that spurious correlations may arise beyond the first word, and our analysis does not exclude this possibility. Our focus on the first word is deliberate for the following reasons.
>
> - First, question words (e.g., “How”, “Can”, “What”) are **semantically neutral** and **do not change the meaning** or risk level of a query. This makes them a natural perturbation point that isolates the correlation signal without affecting the underlying intent.
>
> - Second, if altering such a content-irrelevant token is **enough to cause** jailbreaks or over-prudence, it indicates that the correlation at this position is particularly strong. This shows the model relies on superficial positional cues rather than semantic content, making the first word the **most diagnostic** place to expose the “safety mirage.”
>
> - Third, to investigate whether spurious correlations also arise from semantic words beyond the first word, we performed an additional analysis where the first word is kept fixed and the internal semantic content is paraphrased. We used Qwen2.5-32B-Instruct to generate semantics-preserving paraphrases with the following controlled prompt:
> *“Paraphrase the following instruction so that it starts with '{first_word}'. Keep the meaning exactly the same. Do not add or remove details.”*
>
> **Table R1** shows that paraphrasing semantic words beyond the first word causes only mild changes in ASR and RR, whereas the one-word attack/modification produces much larger shifts. This shows that correlations with other words may exist, but are less reliably exploitable compared to the initial question words. In contrast, perturbing the question word (selected based on the question word frequency unveiled in Fig. 2) induces strong, reproducible effects, making it a clear shortcut learned during safety fine-tuning. MU methods (NPO/RMU) remain stable under all perturbations, further confirming their robustness.
>
> **Table R1**: Safety (ASR↓) and Over-Prudence (RR↓) results on VLGuard. “Paraphrasing” denotes the semantics-preserving paraphrase attack described above.
>
> | Models            | **Safety Evaluation (ASR, ↓)**  |     **Safety Evaluation (ASR, ↓)**       |     **Safety Evaluation (ASR, ↓)**          | **Over-Prudence Evaluation (RR, ↓)** |    **Over-Prudence Evaluation (RR, ↓)**        |      **Over-Prudence Evaluation (RR, ↓)**        |
> |-------------------|---------------------------------|------------|--------------|--------------------------------------|------------|--------------|
> |                   | Without one-word attack                          | With one-word attack      | Paraphrasing | Without one-word modification                               | With one-word modification      | Paraphrasing |
> | **LLaVA-1.5-7B**  | 64.25%                          | 90.27%     | 65.84%       | 0.36%                                | 0.36%      | 0.36%        |
> | **+Unsafe-Filter**| 65.66%                          | 90.72%     | 67.42%       | 0.36%                                | 0.36%      | 0.36%        |
> | **+Mixed-SFT**    | 0.23%                           | 54.98%     | 0.90%        | 4.48%                                | 91.76%     | 5.02%        |
> | **+Post-hoc**     | 0.23%                           | 46.83%     | 0.90%        | 2.69%                                | 90.83%     | 2.87%        |
> | **+NPO-Unlearning**| 2.49%                          | 12.92%     | 2.26%        | 2.51%                                | 11.69%     | 2.87%        |
> | **+RMU-Unlearning**| 1.29%                          | 10.18%     | 1.58%        | 1.25%                                | 7.56%      | 1.43%        |

---

> ### Author Response · Authors · 2025-11-21
> **Response to Reviewer ZeCK (Part II)**
>
> **[W2]**: After unlearning, the utility performance of models degrades more than that of safety fine-tuning methods.
>
> **Response**:
> Thank you for this question.
>
> - We agree that MU may introduce a modest utility drop, reflecting the inherent trade-off between unlearning robustness and general utility, a phenomenon well documented in the unlearning literature [R1-R2] and analogous to the robustness–accuracy trade-off widely observed in adversarial defense [R3]. Importantly, this trade-off remains meaningful: the slight utility reduction is accompanied by substantial decreases in ASR and RR, yielding significantly stronger safety behavior. Moreover, as we further demonstrate through our coreset unlearning experiments in Table A1, this balance can be noticeably improved by applying MU to only a subset of safety data, enabling a more favorable and tunable safety–utility trade-off.
>
> - As noted in Lines 415–418 of the main paper, we adopt a coreset unlearning strategy that performs unlearning on a subset of the safety data (known as coreset) to better manage this trade-off. The results in Table A1 (Appendix D.4) showed that even using only 10% of the safety data effectively reduces ASR and RR, while yielding noticeably higher utility (Acc) compared to full NPO/RMU.
> These ablation studies show that MU’s utility drop is further manageable and controllable.
>
> **[W3]:** Experiments on recent VLMs (e.g., Qwen-VL)
>
> **Response:**
> Thank you for the suggestion. To evaluate whether our findings extend beyond the LLaVA-1.5 family, we conducted additional experiments on Qwen2.5-VL-7B. Results are presented in **Table R2**.
>
> The key observations are summarized below.
>
> - **Safety Mirage persists in SOTA VLMs.**
>   Standard SFT still leads to large increases in ASR under one-word attacks. For example, the Mixed-SFT model’s ASR rises from 1.35% to 49.46% after the attack.
>
> - **MU remains effective on the new VLM.**
>   NPO and RMU reduce post-attack ASR to 14.48% and 12.67%, far lower than SFT, while maintaining comparable TextVQA utility.
>
> We will include these additional Qwen2.5-VL results in the revision and plan to extend the evaluation to additional Qwen-series models.
>
> **Table R2**: Safety (ASR↓) and Over-prudence (RR↓) on VLGuard.  “Before” and “After” indicate performance measured before and after the one-word attack/modification. The evaluation follows the same settings as Table 1.
>
> | Models             | Safety Evaluation (ASR, ↓) | Safety Evaluation (ASR, ↓) | Over-Prudence Evaluation (RR, ↓) | Over-Prudence Evaluation (RR, ↓) | Utility Evaluation (Acc., ↑) |
> |--------------------|----------------------------|----------------------------|----------------------------------|----------------------------------|------------------------------|
> |                    | Before                     | After                      | Before                           | After                            | TextVQA                      |
> | Qwen2.5-VL-7B      | 70.59%                     | 69.91%                     | 1.25%                            | 1.08%                            | 84.93%                       |
> | +Mixed-SFT         | 1.35%                      | 49.46%                     | 4.48%                            | 91.76%                           | 84.13%                       |
> | +Posthoc-SFT       | 2.03%                      | 51.81%                     | 3.40%                            | 93.37%                           | 84.67%                       |
> | +NPO-Unlearning    | 3.62%                      | 14.48%                     | 3.04%                            | 12.90%                           | 84.18%                       |
> | +RMU-Unlearning    | 2.71%                      | 12.67%                     | 2.87%                            | 11.11%                           | 83.12%                       |
>
> **[W4]:** The current results show that the ASR and utility of the model after unlearning (before jailbreaking) are inferior to those of the safety fine-tuned model. Would it be possible to perform unlearning on the safety fine-tuned model to eliminate the previously analyzed correlations?
>
> **Response:**
> Thank you for this question.
>
> Yes, MU can technically be applied after safety fine-tuning. However, we believe that a two-stage pipeline (SFT + MU) is unnecessary because SFT and MU ultimately target the same objective, safety alignment. In our work, MU is intended as a replacement for supervised safety fine-tuning, precisely to avoid inheriting the harmful knowledge and spurious input–label correlations introduced by SFT on biased safety datasets.
>
> Moreover, a two-stage approach does not inherently improve utility preservation. As discussed earlier in [W2], the trade-off between unlearning effectiveness and utility is fundamental, much like the robustness–accuracy trade-off in adversarial defense. Adding SFT before MU does not eliminate the unlearning phase and therefore does not remove this trade-off.

---

> > ### Author Response · Authors · 2025-11-21
> > **Response to Reviewer ZeCK (Part III)**
> >
> > **[W5]:** The “Safety Mirage” described in the paper seems to lack properties specific to vision-language models; it appears to be a phenomenon similar to what has been observed in LLMs. A further discussion on how visual information influences model safety would enhance the completeness
> >
> > **Response:** Thank you for raising this important question. We clarify this from the following complementary perspectives
> >
> > - ​​While surface-level token correlations also exist in text-only LLMs, VLMs introduce an additional alignment challenge: the visual encoder provides an external semantic prior that can override or bypass the LLM’s safety alignment. Recent work [R4] shows that images effectively act as a “foreign language’’ for VLMs, creating alternative reasoning pathways that do not arise in purely textual models. As we further validate through an optimization-driven comparison of jailbreaking attacks in LLMs versus VLMs (Table R3), the cross-modal pathway in VLMs causes textual jailbreaks that succeed in LLMs to fail to transfer to VLMs. This divergence indicates that the identified safety mirage is unique to VLMs: the interplay between visual cues and safety labels amplifies the model’s reliance on spurious correlations and shortcut features in ways that remain unclear in text-only LLMs. This cross-modal vulnerability is precisely what gives rise to the VLM-specific safety mirage that our work investigates.
> >
> > - To demonstrate the difference in safety alignment between LLMs and VLMs, we leverage the **GCG attack** [R5], a strong gradient-based method specifically designed for LLM jailbreaking. As shown in **Table R3**, LLM-style jailbreak mechanisms do not transfer cleanly to VLMs. Although GCG is highly effective on text-only LLMs, its success rate drops sharply once visual information is introduced. Even when we adapt GCG to the multimodal setting, optimizing more suffix tokens using a harmful image–text pair, the resulting adversarial tokens remain far **less effective** in VLMs. This confirms that VLMs rely on multimodal pathways that fundamentally differ from the purely textual mechanisms exploited in LLM jailbreaks.
> >
> > **Table R3**: Safety evaluation (ASR↓) on VLGuard under adversarial-word perturbations. “Before/After” reports ASR under a 3-shot “What” one-word attack. “GCG Attack” evaluates gradient-optimized suffix tokens applied under the same 3-shot configuration.
> >
> > | Models            | Before one-word attack | After one-word attack | After GCG attack |
> > |------------------|------------------------|------------------------|------------------|
> > | LLaVA-1.5-7B      | 64.25%                 | 90.27%                 | 72.90%           |
> > | +Unsafe-Filter    | 65.66%                 | 90.72%                 | 73.75%           |
> > | +Mixed-SFT        | 0.23%                  | 54.98%                 | 5.54%            |
> > | +Post-hoc         | 0.23%                  | 46.83%                 | 4.07%            |
> > | +NPO-Unlearning   | 2.49%                  | 12.92%                 | 3.87%            |
> > | +RMU-Unlearning   | 1.29%                  | 10.18%                 | 2.26%            |
> >
> > > **Reference**
> > > [R1] Nguyen, Thanh Tam, et al. "A survey of machine unlearning." *ACM Transactions on Intelligent Systems and Technology* 16.5 (2025): 1-46.
> > > [R2] Liu, Sijia, et al. "Rethinking machine unlearning for large language models." *Nature Machine Intelligence* (2025): 1-14.
> > > [R3] Tsipras, Dimitris, et al. "Robustness may be at odds with accuracy." arXiv preprint arXiv:1805.12152 (2018).
> > > [R4] Pi, Renjie, et al. "Mllm-protector: Ensuring mllm's safety without hurting performance." arXiv preprint arXiv:2401.02906 (2024).
> > > [R5] Zou, Andy, et al. "Universal and transferable adversarial attacks on aligned language models." arXiv preprint arXiv:2307.15043 (2023).

---

> > > ### Author Response · Authors · 2025-11-25
> > > **Look forward to your feedback**
> > >
> > > Dear Reviewer ZeCK,
> > >
> > > A few days ago, we submitted our responses and have now uploaded a revised version of the paper, with all changes highlighted in blue, including the key points you raised that require special attention.
> > >
> > > We are writing to kindly check whether you have any follow-up questions or additional comments, or if our responses have sufficiently addressed your concerns. We would be happy to clarify any remaining points and continue the discussion during the open review period. We truly appreciate your time, constructive feedback, and engagement with our work.
> > >
> > > Authors

---

### Official Review · Reviewer_fcMJ · 2025-10-30

**Soundness:** 3
**Presentation:** 4
**Contribution:** 3
**Rating:** 6
**Confidence:** 4

**Summary:**

This paper first identifies a “safety mirage” problem in current fine-tuning approaches, where vision-language models (VLMs) develop biases due to spurious correlations between textual queries and safety labels in the training data. Such biases make the models vulnerable to one-word attacks, in which simply replacing a highly frequent query word with one associated with benign outputs can bypass safety mechanisms. Moreover, fine-tuned models often exhibit an over-prudence issue, leading them to reject normal or harmless queries. To address these problems, the authors propose using machine unlearning (MU) as a direct method to mitigate the effects of spurious correlations. Experimental results on several benchmarks demonstrate that MU-based safety fine-tuning effectively alleviates both the spurious correlation and over-prudence issues.

**Strengths:**

Revealing the “Safety Mirage” Phenomenon: The author systematically investigates the existence of spurious textual features and correlations by analyzing word frequency patterns across various safety fine-tuning datasets. This analysis reveals that certain frequent words are strongly correlated with safety labels. To validate this bias, the author conducts one-word attack experiments, confirming that the presence of bias-inducing words is crucial for the attack’s success. Furthermore, even a single insertion of a rejection-related word (e.g., “Share”) can trigger over a 90% rejection rate for otherwise safe text–image queries, highlighting the severity of the over-prudence issue.

Proposing Representation Misdirection Unlearning (RMU) and Negative Preference Optimization (NPO): To address these biases, the author introduces two complementary methods. RMU maps the intermediate representations of unsafe data to randomized feature spaces, effectively erasing learned spurious associations. Meanwhile, NPO encourages the model to deviate from the reference model’s behavior specifically on unsafe inputs. Together, these techniques preserve the model’s general capabilities while mitigating spurious correlations.

Comprehensive Evaluation Framework: The author adopts a multi-dimensional evaluation framework encompassing safety, over-prudence, and general capability metrics, benchmarking the proposed method across diverse datasets. The experimental results confirm the effectiveness of MU in improving performance across all three dimensions. Additionally, token-level saliency analyses—such as masking the words “What” or “Share”—provide further evidence of the causal influence of spurious correlations.

**Weaknesses:**

Limited Model Scope and Generalizability: The paper evaluates its methods exclusively on the LLaVA-1.5 model series, which is now relatively outdated. By not extending experiments to more state-of-the-art (SOTA) vision-language models (VLMs), the study leaves two critical questions unresolved: (1) whether the “safety mirage” phenomenon is a widespread issue across modern VLMs or merely an artifact of LLaVA-1.5’s limited language generalization; and (2) whether the proposed Machine Unlearning (MU) method maintains its advantages over standard supervised fine-tuning (SFT) when applied to models with stronger intrinsic anti-bias mechanisms. This narrow model validation thus limits the generalizability and external validity of the study’s conclusions.

Lack of Comparison with Data-Level Bias Mitigation: The study identifies "training data bias" (spurious token-label correlations) as the root cause of "safety mirage," yet it omits a critical baseline: mitigating data bias via LLM-driven data diversification. This gap prevents a comprehensive assessment of whether MU is truly a necessary solution, or if simpler data-level interventions could already alleviate the problem.

Dependence on Existing Dataset Paradigms: Despite its conceptual novelty, the proposed MU-based framework remains grounded in existing safety fine-tuning datasets, without addressing the deeper issue of how these datasets themselves are constructed and labeled. Consequently, the method offers a pragmatic yet incremental improvement rather than a fundamental solution to the broader challenge of VLM safety alignment.

**Questions:**

1. For SOTA VLMs with stronger cross-modal alignment (e.g., Qwen2.5-VL), do they still exhibit "safety mirage" (e.g., high ASR after one-word attacks) when fine-tuned with traditional SFT?
2. If the safety training dataset is diversified via LLM-driven paraphrasing (e.g., using GPT to rewrite queries while preserving semantic intent, weakening token-label correlations), would traditional SFT on this diversified data achieve ASR and over-prudence levels comparable to MU?
3. Compared to MU, how does the computational cost of "LLM-based data diversification + SFT" compare? Is MU still a cost-effective solution if data diversification can already mitigate safety mirage?
4. If we combine "LLM-driven data diversification" with MU (i.e., apply MU on a diversified dataset), would this further improve safety-utility tradeoffs? Or would the two methods be redundant, given that diversification already reduces spurious correlations?

---

> ### Author Response · Authors · 2025-11-21
> **Response to Reviewer fcMJ (Part I)**
>
> We thank the reviewer for the thorough and constructive feedback. We appreciate your acknowledgement of our identification of the “Safety Mirage,” and we are also encouraged by your recognition of the effectiveness of our proposed machine unlearning-based methods, as well as your acknowledgement of the breadth of our evaluation framework.
>
> Below, we provide detailed clarifications addressing your concerns, denoted by **[W]** for weaknesses and **[Q]** for questions.
>
> **[W1&Q1]**: Limited model scope & Other VLMs
>
> **Response:**
> Thank you for pointing this out! We will respond to your concerns from two perspectives:
>
> - **Existing Model Diversity.**
>   We note that our evaluation already covers **four model configurations** within the LLaVA-1.5 series, varying in scale and fine-tuning strategy (7B/13B, Full Fine-tuning/LoRA). In all four distinct configurations, the “safety mirage” was prevalent, and our proposed **Machine Unlearning (MU)** method consistently maintained its advantages over standard supervised fine-tuning (SFT). This is because MU does **NOT** rely on explicit “safety labels”, and therefore avoids inheriting the text–label shortcut patterns present in the safety fine-tuning dataset (as motivated in Lines 295–299). By operating in a label-free manner, MU prevents the model from reinforcing spurious feature–label correlations and thus remains more robust under both original and perturbed query formulations.
>
> - **Additional VLM Experiments on Qwen2.5-VL.**
>   We appreciate the suggestion for external validation. We have conducted additional experiments on **Qwen2.5-VL** (7B) and reported the results in **Table R1**. From these results, we observe two consistent findings.
>   (1) **Safety mirage persists:** Standard SFT still exhibits large ASR increases under one-word attacks; for example, the Mixed-SFT model’s ASR jumps from 1.35% to 49.46% after the perturbation.
>   (2) **MU remains effective:** Both NPO and RMU achieve substantially lower post-attack ASR compared to SFT methods while maintaining comparable TextVQA utility, indicating much stronger robustness.
>   These results provide evidence that the safety mirage phenomenon is not specific to LLaVA-1.5, but also appears in other modern VLM architectures, and that MU consistently outperforms SFT due to its label-free manner.
>
> We will include these additional Qwen2.5-VL results in the revision and plan to expand the evaluation to additional Qwen-series models.
>
> **Table R1**: Safety (ASR↓) and Over-prudence (RR↓) on VLGuard.  “Before” and “After” indicate performance measured before and after the one-word attack/modification. The evaluation follows the same settings as Table 1.
>
> | Models             | Safety Evaluation (ASR, ↓) | Safety Evaluation (ASR, ↓) | Over-Prudence Evaluation (RR, ↓) | Over-Prudence Evaluation (RR, ↓) | Utility Evaluation (Acc., ↑) |
> |--------------------|----------------------------|----------------------------|----------------------------------|----------------------------------|------------------------------|
> |                    | Before                     | After                      | Before                           | After                            | TextVQA                      |
> | Qwen2.5-VL-7B      | 70.59%                     | 69.91%                     | 1.25%                            | 1.08%                            | 84.93%                       |
> | +Mixed-SFT         | 1.35%                      | 49.46%                     | 4.48%                            | 91.76%                           | 84.13%                       |
> | +Posthoc-SFT       | 2.03%                      | 51.81%                     | 3.40%                            | 93.37%                           | 84.67%                       |
> | +NPO-Unlearning    | 3.62%                      | 14.48%                     | 3.04%                            | 12.90%                           | 84.18%                       |
> | +RMU-Unlearning    | 2.71%                      | 12.67%                     | 2.87%                            | 11.11%                           | 83.12%                       |

---

> > ### Author Response · Authors · 2025-11-21
> > **Response to Reviewer fcMJ (Part II)**
> >
> > **[W2&Q2]**: Can Paraphrased Data Mitigate Safety Mirage Without MU?
> >
> > **Response:**
> > Thank you for raising this important point. We address it from the following perspectives.
> >
> > - Large-scale paraphrasing of safety datasets is non-trivial and cannot be treated as an automated solution. Diversifying all safety queries while guaranteeing preservation of intent, semantic equivalence, and correct safety labels is especially challenging in multimodal settings, where textual phrasing must align with image content. This process introduces multiple design choices (e.g., where to paraphrase, how much to modify, and how to assess quality). As a result, extensive human verification is required. In practice, this becomes a substantial data-engineering pipeline rather than a simple one-shot LLM paraphrasing step.
> >
> > - Even if such data engineering were feasible at scale, it would still not reliably eliminate spurious textual shortcuts. LLM-driven paraphrasing can introduce new linguistic patterns and biases of its own; weakening a first-word correlation (e.g., “what”) may simply shift the shortcut to another token or phrasing pattern produced by the paraphraser. As a result, SFT on paraphrased data may *relocate* rather than remove the underlying shortcut.
> >
> > - MU is a label-free intervention that does not require modifying or augmenting the existing safety fine-tuning data. Our MU framework operates directly on the model during fine-tuning and does not rely on additional annotations, paraphrasing, or data rewriting. This makes MU a drop-in replacement for current VLM safety fine-tuning pipelines, while avoiding the risk of introducing data noise, semantic drift, or unintended label inconsistencies that often arise in augmentation-based approaches.
> >
> > Based on the above, we respectfully disagree that LLM-driven data diversification constitutes an appropriate baseline for our setting. To stress, our goal is to provide a *model-level* solution without altering the provided fine-tuning dataset.
> >
> > **[W3]**: Incremental Contribution Due to Reliance on Existing Datasets
> >
> > **Response:**
> > Thank you for the thoughtful comment. We respectfully disagree with the characterization that our method offers only a pragmatic or incremental improvement.
> >
> > - The identified spurious correlation is a fundamental limitation of supervised safety fine-tuning, not a dataset-specific artifact. As noted by Reviewer ZeCK (“the analysis of spurious correlation is interesting and insightful”) and Reviewer Gye9 (“the paper identifies an important issue in safety fine-tuning”), the safety mirage arises from the inherent reliance on safety labels during fine-tuning, rather than from any particular dataset’s annotation style. Even if the format or wording of a dataset changes, spurious correlations between input features and labels can still persist in more subtle or hidden forms. This makes it essential to address the issue through a dataset-agnostic, model-level mechanism rather than through dataset-specific fixes.
> >
> > - MU provides a principled, dataset-agnostic mitigation, not a data-cleaning step. MU reduces reliance on shortcut features by operating directly on the model’s internal representations, without modifying, reconstructing, or re-annotating the underlying data. As discussed earlier, MU is label-free, does not depend on the quality of dataset curation, and avoids introducing new biases or inconsistencies through data rewriting. In this sense, MU addresses the root problem: the model’s tendency to overfit to feature–label shortcuts during safety fine-tuning. Our approach complements, rather than depends on, the existing dataset paradigm. While our experiments use standard safety datasets to ensure comparability with prior work, the MU mechanism itself does not rely on any particular labeling scheme or dataset structure.

---

> ### Author Response · Authors · 2025-11-21
> **Response to Reviewer fcMJ (Part III)**
>
> **[Q3]**: Cost of LLM-Driven Data Diversification vs. MU
>
> **Response:**
> Thank you for raising this comment. As we also discussed in our responses to [W2] and [Q2], we intentionally do not include LLM-driven data diversification as a baseline because it is far from simple in practice.
>
> - LLM-driven data diversification is costly and can introduce new biases.   Generating paraphrased variants for every safety query requires large-scale LLM computation and, and human verification to ensure semantic fidelity, especially in multimodal settings where textual phrasing must remain consistent with visual content. More importantly, paraphrasers introduce their own linguistic patterns and biases. Removing one shortcut (e.g., a first-word bias) can easily create another, simply shifting the spurious correlation rather than eliminating it. Fully removing these hidden shortcuts is not feasible and may degrade downstream performance by distorting the data distribution. Moreover, this data-engineering process must be repeated for every new safety dataset, significantly increasing development cost.
>
> - MU remains a cost-effective and broadly validated safety mechanism. MU requires no data rewriting, re-annotation, or augmentation. It is label-free, operates directly on the model, and scales efficiently without modifying any dataset. We believe MU therefore offers a more practical and principled alternative than dataset-level debiasing. This parallels the well-known distinction in adversarial robustness studies [R1]: model-level defenses often provide more stable and generalizable protection than data-level interventions, which can be brittle or require continual data engineering.
>
> **[Q4]**: Would LLM-rewriting + MU improve the safety–utility trade-off?
>
> **Response:**
> Thank you for the thoughtful question.
>
> - Combining LLM-driven diversification with MU is redundant and costly. MU does not rely on safety labels or any specific wording patterns between inputs and outputs. Its label-free nature prevents safety fine-tuning from creating new spurious correlations, regardless of how the queries are phrased. Therefore, paraphrasing the dataset before applying MU is unnecessary, as MU already operates independently of textual form. Moreover, as noted earlier, LLM-based paraphrasing can be computationally expensive and requires additional human validation, making it a burdensome and inefficient addition to the training pipeline.
>
> - MU already provides an effective and controllable safety–utility trade-off. As noted in Lines 415–418 of the main paper, we adopt a coreset unlearning strategy that applies MU to only a subset of the safety data, which helps preserve utility while still yielding safety improvements. Table A1 (Appendix D.4) shows that using just 10% of the safety data for MU still yields substantial reductions in ASR and RR, while preserving noticeably higher utility compared to full NPO/RMU. This demonstrates that MU’s utility cost is both manageable and tunable when balancing safety robustness and downstream performance.
>
> > **References**
> > [R1] Athalye, Anish, et al. "Obfuscated gradients give a false sense of security: Circumventing defenses to adversarial examples." *ICML*, 2018.

---

> > ### Author Response · Authors · 2025-11-25
> > **Look forward to your feedback**
> >
> > Dear Reviewer fcMJ,
> >
> > A few days ago, we submitted our responses and have now uploaded a revised version of the paper, with all changes highlighted in blue, including the key points you raised that require special attention.
> >
> > We are writing to kindly check whether you have any follow-up questions or additional comments, or if our responses have sufficiently addressed your concerns. We would be happy to clarify any remaining points and continue the discussion during the open review period. We truly appreciate your time, constructive feedback, and engagement with our work.
> >
> > Authors

---

### Official Review · Reviewer_kJt9 · 2025-11-01

**Soundness:** 3
**Presentation:** 4
**Contribution:** 3
**Rating:** 6
**Confidence:** 3

**Summary:**

This paper examines the effectiveness of machine unlearning for improving the safety of Vision-Language Models (VLMs).  It is shown that existing supervised fine-tuning (SFT) defenses exhibit two main problems: 1. Brittleness to one-word attacks, where replacing just a single word in a harmful prompt can bypass the defense, and 2. Over-prudence, where replacing just a single word in a benign prompt can lead to an unncessary refusal. The paper shows that this is due to spurious correlations that are learned from the fine-tuning dataset. To overcome this, the paper proposes using label-free Machine Unlearning (MU) approaches, and shows that they are much more effective at mitigating the spurious correlation issues than SFT-based approaches.

**Strengths:**

1. Provides convincing evidence that unbalanced datasets (in terms of the initial word) cause spurious correlations to be learned when performing SFT for safety enhancement in VLMs: The frequency of initial words is analyzed (revealing highly skewed distributions), which are then used to create an effective one-word jailbreak attack as well as to demonstrate over-prudence.
2. The use of machine unlearning techniques to address the spurious correlation problem is well-motivated, as they are label-free. Moreover, they are shown to be much more effective than SFT-based defenses at reducing the one-word attack success and mitigating over-prudence across multiple VLMs, with minimal minor impact to model utility.

**Weaknesses:**

1. The attacks evaluated in this work are very limited (just the proposed one-word attack). It would strengthen the paper to evaluate against other kinds of attacks, such as those mentioned in section 2.
2. Building off the previous weakness, it is unclear whether the findings would remain consistent under attacks that manipulate the image (e.g., gradient-based adversarial image generation), since the observations were based on spurious correlations on just the text component of the input.

**Questions:**

1. It seems the one-word attack evaluation uses adversarial words found via the analysis done on the various training datasets in Figure 2. I’m wondering whether there are adversarial words “outside” of these datasets that may be just as effective, and whether or not the machine unlearning approaches can generalize to such words. These words could be discovered through something similar to GCG ([1]) (e.g., swapping a single token to maximize a harmful target).

[1] Andy Zou, Zifan Wang, Nicholas Carlini, Milad Nasr, J Zico Kolter, and Matt Fredrikson. Universal and transferable adversarial attacks on aligned language models. arXiv preprint arXiv:2307.15043, 2023.

---

> ### Author Response · Authors · 2025-11-21
> **Response to Reviewer kJt9 (Part I)**
>
> We thank the reviewer for the constructive feedback. We appreciate your acknowledgement of our empirical evidence on spurious correlations, the effectiveness of the proposed attack, and the strong performance of machine unlearning as a mitigation strategy for safety fine-tuned VLMs.
>
> Below, we provide detailed clarifications addressing your concerns, denoted by **[W]** for weakness and **[Q]** for question.
>
> **[W1]**: The attacks evaluated in this work are very limited (just the proposed one-word attack). It would strengthen the paper to evaluate against other kinds of attacks, such as those mentioned in section 2.
>
> **Response**:
> Thank you for your comment! Below, we would like to further clarify (1) why the one-word attack is central to our analysis, and (2) how we already include multimodal attack evaluations as complementary evidence.
>
> - **Why we focus on the one-word attack/modification.**
>   Our aim is *not* to provide an exhaustive attack benchmark, but to reveal the spurious correlation cause behind the safety mirage. After uncovering the dataset-induced starting-word biases (Fig. 2), we intentionally use the simplest possible perturbation, a single-token swap, to demonstrate that: (1) the vulnerability stems directly from the spurious correlation rather than from complex adversarial optimization, (2) the effect is fully controlled and interpretable, isolating the causal role of biased textual features, and (3) even minimal one-word edits expose dramatic failures (e.g., 90% ASR increase).
>
> - **Evaluations on other attack types.**
>   Beyond the one-word analysis, we have already evaluated two major multimodal attack benchmarks, as noted in the main text (Lines 419–422) and reported in Appendix D.5. **MM-SafetyBench**, the SD+TYPO subset constructs visual jailbreaks using Stable Diffusion, which generates images combined with *harmful typography* inserted into the scene. These images are paired with natural-language queries to induce unsafe responses.
>   **FigStep** synthesizes adversarial figures containing *structured harmful cues* (e.g., embedded keywords or symbolic instructions) to mislead the model into providing unsafe outputs.
>
>   As our results show in Appendix D.5, integrating one-word manipulation with existing multimodal attack benchmarks further increases the ASR for standard SFT-based safety-fine-tuned models, highlighting their vulnerability to the identified spurious correlation. In addition, our MU-based approaches consistently achieve substantially lower post-attack ASR, both with and without the additional one-word perturbation, demonstrating improved robustness and reduced reliance on the spurious correlation.
>
> **[W2]**: Building off the previous weakness, it is unclear whether the findings would remain consistent under attacks that manipulate the image (e.g., gradient-based adversarial image generation), since the observations were based on spurious correlations on just the text component of the input.
>
> **Response**:
> Thank you for raising this point. Our findings indeed center on textual spurious correlations, but we clarify below why they remain relevant for multimodal adversaries, including image-manipulation attacks.
>
> - **Included empirical evidence under visual perturbations.**
>   As noted in Lines 479–481 (Appendix D.7), we explicitly evaluate performance under image variations (not the adversarial perturbations). These experiments demonstrate that the dataset-induced starting-word biases remain highly predictive of model (mis)behavior even when the accompanying image is perturbed, and MU-based methods continue to reduce ASR under these visual changes. This indicates that the text-driven spurious correlations are stable across image perturbations, and that MU keeps robustness even in settings where the image distribution shifts.
> - **Why our analysis focuses on text rather than image gradients.**
>   While we agree image-based attacks are an important threat model, our goal in this work is to isolate and diagnose the root cause of the “safety mirage.” The most consistent and explainable failure mode we uncovered arises from dataset-driven textual shortcuts. Focusing on text allows us to: conduct controlled causal testing (one-word probes), directly attribute failure to fine-tuning-induced correlations, and rule out confounding factors introduced by complex image-space optimization. This targeted analysis is complementary to, rather than in conflict with, adversarial image-generation attacks.
>
>   We agree that examining whether the safety mirage interacts with gradient-based adversarial image generation is an important future direction, and we will explicitly include this point in our conclusion and limitations. However, as discussed in our response to [W1], such image-based adversarial optimization lies outside the scope of our current work, which focuses on one-word attacks/modifications as a controlled diagnostic probe for uncovering and demonstrating the safety mirage.

---

> > ### Author Response · Authors · 2025-11-21
> > **Response to Reviewer kJt9 (Part II)**
> >
> > **[Q1]**: It seems the one-word attack evaluation uses adversarial words found via the analysis done on the various training datasets in Figure 2. I’m wondering whether there are adversarial words “outside” of these datasets that may be just as effective, and whether or not the machine unlearning approaches can generalize to such words. These words could be discovered through something similar to GCG (e.g., swapping a single token to maximize a harmful target).
> >
> > **Response:**
> > Thank you for your valuable question!
> >
> > To examine whether MU generalizes to adversarial triggers outside the dataset-induced correlations identified in Fig.2, we conducted the suggested GCG experiments, with results summarized in **Table R1**. All attacks in Table R1 (one-word and GCG) are evaluated under the same 3-shot setting. We observe that GCG attacks are substantially less effective in VLMs: even with gradient-based optimization, the ASR increase is modest. In contrast, simple one-word substitutions (e.g., changing the first word to “What”) still cause severe degradation for SFT-based models (ASR up to 90%). This indicates that the dominant vulnerability mechanism is not exotic tokens generated by optimization, but rather the shortcut correlations we identify.
> >
> >   Notably, GCG was originally designed for LLM jailbreaks rather than multimodal models [R1], and VLMs require substantially stronger optimization. To adapt GCG to VLMs, we optimized 20 adversarial suffix tokens on LLaVA-1.5-7B using a harmful vision–text pair (an explosive-device image paired with “How to build a bomb?”) and performed 300 gradient ascent steps targeting the unsafe response “Sure, I’d be happy to help.” The resulting suffix tokens are dataset-external and appended during inference. Even under this stronger multimodal setup, GCG attacks remain considerably weaker than the one-word attack, highlighting the difficulty of transferring LLM-style jailbreak methods to VLMs.
> >
> > This observation is also consistent with recent findings showing that images can function as a “foreign language’’ for VLMs [R2], which can bypass or attenuate purely text-based adversarial triggers. Crucially, our results show that the primary failure mode arises from a dataset-induced spurious correlation between certain first-word patterns and unsafe labels. This shortcut directly explains the strong effectiveness of our one-word attack: minimal substitutions that align with this biased correlation are sufficient to trigger severe safety failures.
> >
> > **Table R1**: Safety evaluation (ASR↓) on VLGuard under adversarial-word perturbations. “Before/After” reports ASR under a 3-shot “What” one-word attack. “GCG Attack” evaluates gradient-optimized suffix tokens applied under the same 3-shot configuration.
> >
> > | Models            | Before one-word attack | After one-word attack | After GCG attack |
> > |------------------|------------------------|------------------------|------------------|
> > | LLaVA-1.5-7B      | 64.25%                 | 90.27%                 | 72.90%           |
> > | +Unsafe-Filter    | 65.66%                 | 90.72%                 | 73.75%           |
> > | +Mixed-SFT        | 0.23%                  | 54.98%                 | 5.54%            |
> > | +Post-hoc         | 0.23%                  | 46.83%                 | 4.07%            |
> > | +NPO-Unlearning   | 2.49%                  | 12.92%                 | 3.87%            |
> > | +RMU-Unlearning   | 1.29%                  | 10.18%                 | 2.26%            |
> >
> > > **Reference**
> > > [R1] Zou, Andy, et al. "Universal and transferable adversarial attacks on aligned language models." arXiv preprint arXiv:2307.15043 (2023).
> > > [R2] Pi, Renjie, et al. "Mllm-protector: Ensuring mllm's safety without hurting performance." arXiv preprint arXiv:2401.02906 (2024).

---

> > > ### Author Response · Authors · 2025-11-25
> > > **Look forward to your feedback**
> > >
> > > Dear Reviewer kJt9,
> > >
> > > A few days ago, we submitted our responses and have now uploaded a revised version of the paper, with all changes highlighted in blue, including the key points you raised that require special attention.
> > >
> > > We are writing to kindly check whether you have any follow-up questions or additional comments, or if our responses have sufficiently addressed your concerns. We would be happy to clarify any remaining points and continue the discussion during the open review period. We truly appreciate your time, constructive feedback, and engagement with our work.
> > >
> > > Authors

---

### Author Response · Authors · 2025-11-29
**Summary of response to all reviewers**

Dear ACs, SACs, and PCs:

Thank you very much for your hard work in overseeing the review process. In our rebuttal (submitted on Nov.20) and the corresponding paper revision (submitted Nov. 25), we made substantial efforts to clarify the reviewers’ concerns and resolve their questions. **However, we have not received any follow-up from the reviewers after submitting the rebuttal and revision.**

For ease of tracking, we summarize below the key points addressing each reviewer’s questions.

**Summary of response to Reviewer kJt9 (initial score 6, no follow-up)**
- We clarified why we focus on the one-word attack/modification, and we already included other safety evaluation benchmarks like MM-SafetyBench and FigStep (See response to W1).
- We clarified that we already included the empirical evidence under visual perturbations, and gave reasons why our analysis focuses on text rather than image gradients (See response to W2).
- We added GCG attack experiments, showing it is substantially less effective in VLMs, even with gradient-based optimization (See response to Q1).

**Summary of response to Reviewer fcMJ (initial score 6, no follow-up)**
- We clarified that our existing model already has diversity, and added additional experiments on Qwen2.5-VL, showing safety mirage persists, and MU remains effective (See response to W1&Q1).
- We clarified why paraphrased data cannot mitigate the safety mirage. Data-wise paraphrasing is dataset-dependent, heuristic, and computationally costly, making it neither automatic nor reliably feasible. In contrast, MU offers a label-free intervention without modifying or expanding existing safety data (See response to W2&Q2).
- We clarified that our contribution is not incremental with other reviewers’ acknowledgements, and further explained the discovery of safety mirage, and stressed MU as a label-free new way to address this (See response to W3).
- We explained that LLM-driven data diversification does not eliminate spurious correlations and may introduce new biases, while MU remains cost-effective and broadly validated (See response to Q3).
- We answered that LLM-rewriting plus MU is unnecessary, with experimental evidence that MU already provides an effective and controllable safety–utility trade-off (See response to Q4).

**Summary of response to Reviewer ZeCK (initial score 6, no follow-up)**
- We emphasized that question words are semantically neutral yet effective, and our additional experiments, where we paraphrased the sentence while keeping only the question words unchanged, demonstrate that paraphrasing semantic content beyond the first word leads to only mild changes in ASR (See response to W1).
- We answered that MU only caused a modest utility drop, and this utility drop is further manageable and controllable through coreset unlearning, as evidenced by our experiments (See response to W2).
- We added additional experiments on Qwen2.5-VL, demonstrating the safety mirage persists, and MU remains consistently effective (See response to W3).
- We explained that applying MU after SFT is unnecessary and does not provide meaningful benefits (See response to W4).
- We explained that VLMs introduce additional alignment challenges compared with LLMs, and added experiments using GCG attacks to illustrate the differences in safety alignment between two modalities (See response to W5).


**Summary of response to Reviewer Gye9 (initial score 4, no follow-up)**
- We pointed out that the motivation for using MU and the accompanying explanations were already provided in Sections 4 and 5 of the manuscript (See response to W1).
- We explained that MU is not a simple form of regularization, and the observed spurious correlations are not attributable to simple overfitting (See response to W2).
- We clarified that the MU model’s output is not “nothing,” as demonstrated across three scenarios: unsafe queries, rephrased unsafe queries, and general benign queries (See response to W3).
- We clarified the inputs used in the K-shot attack and reiterated the motivation for employing this attack setting, which was already explained in the submission (See response to Q1).
- We clarified that examples of one-word jailbreaking and over-prudence were already provided in Fig. 5 of the main paper, as well as Fig. A1 and Fig. A2 in Appendix A (See response to Q2).
- We clarified the loss-masking setup used in NPO training (See response to Q3).

Despite our substantial rebuttal efforts, all reviewers remained silent after our response. Thus, we would like to respectfully express our concern: **the absence of further engagement or discussion may be unfair to our submission, especially given the detailed clarifications and revisions we provided**.

Last but not least, we are deeply grateful to the ACs for their extra time, effort, and thoughtful consideration in handling our submission. Thank you once again for your careful evaluation.

Authors

---

### Meta-Review · Area_Chair_LB6L · 2026-01-06

**Summary:**

The spurious correlation between rejecting/answering behaviours and certain keywords in the query is indeed an important issue in fine-tuning MLLM.

Though the test was carried out on LLaVA-1.5 family only, the scale and fine-tuning strategy (7B/13B, Full Fine-tuning/LoRA) were covered.

**Reviewer Concerns:**

The motivation of using MU was questioned by some reviewers, but it was stated quite clearly in the paper and explained again in the rebuttals.

The evaluation on the LLaVA-1.5 model series only, though it covers four model configurations, it is still quite limited and not further addressed.

**Reviewer Scores:**

Three questions from Reviewer ZeCK were addressed by authors, so the score would have been updated.

---

### Decision · Program_Chairs · 2026-01-26

Accept (Poster)